# FLEXOLMO:
# Open Language Models for Flexible Data Use

**Weijia Shi**[*aw]   **Akshita Bhagia**[*a]   **Kevin Farhat**[*a]   **Niklas Muennighoff**[as]   **Jacob Morrison**[aw]

**Pete Walsh**[a]   **Dustin Schwenk**[a]   **Shayne Longpre**[m]   **Jake Poznanski**[a]

**Allyson Ettinger**[a]   **Daogao Liu**[w]   **Margaret Li**[w]   **Dirk Groeneveld**[a]   **Mike Lewis**[w]

**Wen-tau Yih**[w]   **Luca Soldaini**[a]   **Kyle Lo**[a]   **Noah A. Smith**[a]   **Luke Zettlemoyer**[w]

**Pang Wei Koh**[aw]   **Hannaneh Hajishirzi**[aw]   **Ali Farhadi**[aw]   **Sewon Min**[*ab]

[a]Allen Institute for AI   [w]University of Washington   [b]University of California, Berkeley
[s]Stanford University   [m]MIT

swj0419@uw.edu   akshitab@allenai.org   sewonm@berkeley.edu

🤗 **Model**   hf.co/allenai/FlexOlmo-7x7B-1T
🞛 **Code**   github.com/allenai/FlexOlmo
**Blog**   allenai.org/blog/flexolmo

## Abstract

We introduce **FLEXOLMO**, a new class of language models (LMs) that supports (1) *distributed training without data sharing*, where different model parameters are independently trained on closed datasets, and (2) *data-flexible inference*, where these parameters along with their associated data can be flexibly included or excluded from model inferences with no further training. FLEXOLMO employs a mixture-of-experts (MoE) architecture where each expert is trained independently on closed datasets and later integrated through a new domain-informed routing without any joint training. FLEXOLMO is trained on FLEXMIX, a corpus we curate comprising publicly available datasets alongside seven domain-specific sets, representing realistic approximations of closed sets. We evaluate models with up to 37 billion parameters (20 billion active) on 31 diverse downstream tasks. We show that a general expert trained on public data can be effectively combined with independently trained experts from other data owners, leading to an average 41% relative improvement while allowing users to opt out of certain data based on data licensing or permission requirements. Our approach also outperforms prior model merging methods by 10.1% on average and surpasses the standard MoE trained without data restrictions using the same training FLOPs. Altogether, this research presents a solution for both data owners and researchers in regulated industries with sensitive or protected data. FLEXOLMO enables benefiting from closed data while respecting data owners' preferences by keeping their data local and supporting fine-grained control of data access during inference.

## 1   Introduction

Pretraining language models (LMs) typically requires centralized access to all data during training and does not have any mechanism to track or control the influence of specific data points on model parameters. Model developers must therefore make a one-time decision on which data sources to

---

[*]Core contributors.

39th Conference on Neural Information Processing Systems (NeurIPS 2025).

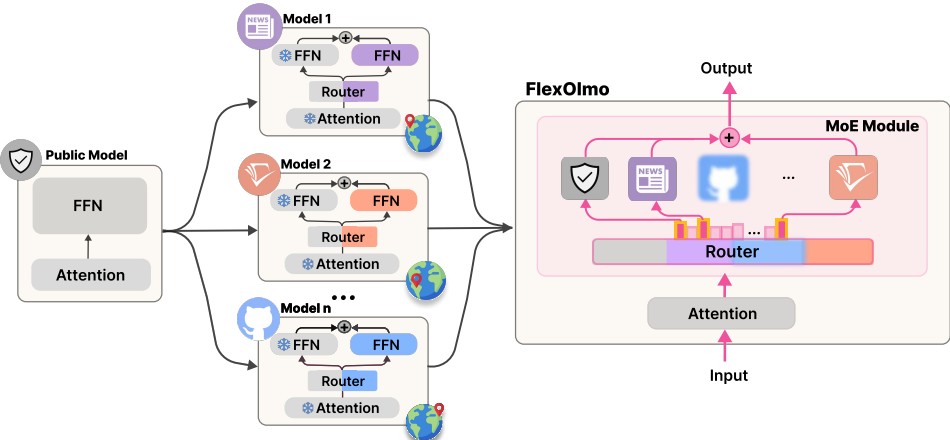

Figure 1: **An overview of FLEXOLMO.** Data owners can contribute without sharing the data by training their own expert modules (FFNs and router embeddings) with a shared public model as an anchor point. At inference, these modules are integrated into a MoE model via a novel router embedding concatenation. This design enables flexible inclusion or exclusion of experts and strict opt-out guarantees, e.g., Github data can be excluded at no cost (blurred) during inference.

include, with limited ability to remove the effect of certain data after training [1, 2, 3]. Moreover, this centralized approach precludes the use of closed data that data owners cannot share with model developers for confidentiality, regulatory, or other reasons. Although solutions have been proposed to allow training without sharing the data, such as federated learning [4, 5], their practical adoption remains limited due to performance degradation and the high cost of synchronized training [6, 7].

We introduce FLEXOLMO, a new class of LMs that enables distributed training on locally maintained datasets while enabling flexible opt-in and opt-out during inference. FLEXOLMO (Figure 1) employs a mixture-of-experts (MoE) architecture [8, 9], where each expert is trained independently on closed datasets and later integrated into an MoE. This design allows data owners to contribute asynchronously without sharing their data, while also enabling continual updates with new data and providing strong guarantees for data opt-out during inference. Our approach can be seen as an instance of model merging [10], which merges different models into a unified one [11, 12]. However, our model is designed to address the unique challenges in our problem setup—combining models pre-trained on completely disjoint datasets with different distributions—which makes prior model merging techniques like ensembling output probabilities [11] or merging model weights [12] suboptimal.

A key challenge in training FLEXOLMO is ensuring the merging of independently trained experts without joint training. We introduce a training algorithm where each data owner independently trains an expert module using the frozen public model as a shared anchor (Figure 1). This approach teaches independently trained experts to coordinate with the same public model and, by extension, with each other. Additionally, the router, a module that determines which experts process each token, typically requires joint training. We address this by assigning each expert a router embedding, initialized from its domain embedding using an off-the-shelf embedder [13] and further finetuned on its corresponding data during individual expert training. These embeddings are then concatenated to form the router during merging, removing the need for joint training.

To validate FLEXOLMO, we curate a data mixture called FLEXMIX, which includes a public training set along with seven domain-specific sets (e.g., news, educational text, and Reddit). These domains are chosen to simulate scenarios where high-quality data that can benefit LM training is not publicly available.

We train FLEXOLMO first on public data, then extend it by merging expert modules trained independently on our simulated closed sets. While continued pretraining on these sets improves some downstream tasks, it suffers from catastrophic forgetting and inconsistent performance. In contrast, FLEXOLMO improves upon the public model by 41% and also outperforms prior merging techniques such as model soup and ensembling by 10.1% across 31 downstream tasks. We observe the largest improvements on tasks related to closed sets. Notably, even on benchmarks where no individual

closed set improved performance over the public model, combining multiple experts yielded significant gains, demonstrating synergies among independently trained modules. Our qualitative analysis demonstrates that sparse expert activation across layers through the MoE architecture is key to these gains. Our qualitative analysis shows that the MoE architecture's ability to selectively activate different experts per layer per token is crucial to these gains by combining the strengths of each specialized expert. We hope our work enables research with a broader range of closed datasets for LM training, particularly for organizations interested in collaborating on scientific research through the new features that FlexOlmo provides.

## 2 Background & Related Work

### 2.1 Background: Data Restrictions

The standard LM training practice requires model developers to aggregate all data centrally and make a one-time decision on which data source to include and exclude. But many real-world data come with sharing and usage restrictions and necessitates (1) model training without data pooling and (2) model inference that can flexibly select different data sources based on use case and access privileges.

**Data Sharing Constraints**    Organizations in regulated industries require LMs that can leverage their closed datasets while maintaining strict data privacy and access controls. Healthcare institutions, financial firms, and other entities possess valuable domain-specific data but cannot share it externally due to HIPAA, GDPR [14, 15], data sovereignty laws [16], and intellectual property (IP) protections. These organizations need training paradigms that enable AI improvement on their sensitive data while ensuring such sensitive data never leaves certain environments and can be removed from the model after training, e.g., when data usage rights expire. In such settings, modular training approaches, where individual experts are trained independently and asynchronously on locally maintained data, are essential.

**Data Use Constraints**    The inclusion of certain data depends on specific use cases and end users. *Privileged access*: User-facing applications often involve closed data restricted to specific, authorized users [17]. For example, GitHub Copilot must tailor code suggestions to reflect internal repositories based on an engineer's role and access rights [18]. *Copyright and data consent*: Legal and ethical considerations on training data for AI are evolving and uncertain [19, 20, 21, 22, 23, 24], and often depend on the data's intended use, e.g., licenses may prohibit commercial use or limit certain query types [25, 26]. *Model control*: Training data often include sensitive content [27, 28, 29] which may be beneficial in certain contexts but harmful in others. For instance, one may want to activate the use of toxic content for toxicity classification in a research setting, but deactivate it in applications presented to a general audience.

### 2.2 Related Work

**Federated Learning**    Federated Learning (FL) trains a single model over distributed datasets by synchronously aggregating client updates [5, 4, 30]. FL methods range from classical approaches that iteratively aggregate parameter updates from local clients [5, 31] to parameter-efficient techniques which have been adapted for LMs [32, 33, 34]. FL can guarantee data privacy using techniques such as homomorphic encryption [35] and differential privacy (DP) [36]. However, FL has seen limited adoption in LM training due to the high cost of synchronization and performance degradation [6, 7], and remains susceptible to privacy attacks due to inter-client communication [37, 38].

Similar to [39], our approach avoids data sharing but differs fundamentally by supporting independent, asynchronous training without costly inter-client communication, and allowing real-time opt-in and opt-out. Like FL, our model allows data owners to optionally apply DP training locally for privacy guarantees. Because DP is orthogonal to our architecture, each contributor can independently choose whether to apply it, providing flexibility without compromising the overall design.

**Model Merging**    Our work builds on recent efforts [40, 10] that advocate for developing machine learning models like open-source softwares, where sub-parts of the model can be trained independently and subsequently merged into unified systems. This can be achieved through various methods, including weight merging, output ensembling, and expert routing. Model soup—merging model weights trained on different datasets from the same initialization—can boost performance [41, 12,

42, 43, 44], especially with weighted combinations [45, 46, 47, 48, 49, 50, 51]. Weighted output ensembling (e.g., BTM [52, 53]) is also effective when models are trained on distinct datasets initialized from the same seed model. These approaches can be applied to our setting, where each expert is independently trained starting from the same public model then merged into a unified one. Our experiments (§5) show that these methods are less effective, primarily because they lack learned connections between different modules, which constrains the expressivity of the resulting models.

An alternative line of work focuses on expert routing methods, such as DEMix [54], BTX [55] and its extensions [56, 57, 58], which merge dense, independently trained experts into a mixture-of-experts (MoE) framework. We draw inspiration from this work, as we also integrate independently trained models into a MoE. However, these methods require joint training on a union of all datasets used in expert training after merging. By contrast, FLEXOLMO removes the need for joint data access to enable training on locally maintained datasets. Our work is also related to ModuleFormer [59], which induces sparse modularity from uncurated data using novel load balancing and concentration losses.

Related efforts in parameter-efficient training have explored merging LoRA adapter weights trained on separate datasets [60, 61, 62, 63], particularly to reduce communication overhead in collaborative settings and support fine-grained data access control and opt-out use cases [64, 65]. Unlike these methods, which focus on merging lightweight adapters, our approach merges full expert models into a standard MoE architecture.

**Mixture-of-Experts (MoE)**   MoE models [8, 9, 66], consisting of many small feedforward networks called *experts*, have gained popularity for their training and inference efficiency. Our work leverages the MoE architecture; however, our motivation and training method are fundamentally different as our primarily goal is to support modularity rather than efficiency.

# 3   FLEXOLMO: LMs with Flexible Data Use

## 3.1   Problem Setup

Let $M_{\text{pub}}$ be a model trained on a publicly available dataset $D_{\text{pub}}$, and $\mathcal{D} = \{D_1, D_2, ..., D_n\}$ represent a collection of locally maintained datasets with separate owners. Our objective is a single model $M_{\text{final}}$, which is constructed via composing $M_{\text{pub}}$ and a set of modules $\{M_1, M_2, \ldots, M_n\}$, where each $M_i$ is independently trained by the owner of $D_i$, who also has access to $M_{\text{pub}}$.

This model satisfies two requirements: (1) training $M_{\text{final}}$ does not require anyone to have joint access to the full dataset collection $\mathcal{D}$, as each $M_i$ is trained independently by the owner of dataset $D_i$; (2) removing any module $M_i$ from $M_{\text{final}}$ guarantees complete removal of its associated data $D_i$.

The key modeling challenges are: (1) to develop an algorithm that creates $M_i$ using $D_i$ and $M_{\text{pub}}$, and (2) to design the merging algorithm that combines $M_{\text{pub}}, M_1, M_2, \ldots, M_n$ into $M_{\text{final}}$.

## 3.2   Model Architecture

FLEXOLMO follows the standard MoE architecture: it replaces the feedforward network (FFN) in each transformer block with a router and $n$ small FFNs called expert modules $\{M_{\text{pub}}, M_1, ..., M_n\}$. Note that we omit the layer index for each expert in our notation for simplicity. Given a processed input token embedding $\mathbf{x} \in \mathbb{R}^h$, the MoE module computes output representation $\mathbf{y}$:

$$\mathbf{y} = \sum_{i \in \text{Top}k(r(\mathbf{x}))} \text{softmax}(r(\mathbf{x})_i) M_i(\mathbf{x}),$$

where the router function $r$ computes the expert probabilities from $\mathbf{x}$. Unlike standard MoEs where experts are trained jointly, our experts are trained asynchronously on distinct datasets $\{D_1, ..., D_n\}$.

## 3.3   Training Algorithm

Standard MoEs train all experts and the router jointly on all data. In contrast, FLEXOLMO trains experts independently by teaching them to coordinate (§3.3.1) and merges them at inference using a domain-informed router (§3.3.2). Optional router tuning can further improve performance (§3.3.3).

### 3.3.1 Training Experts to Coordinate

A straightforward way to train each expert would be to directly continue to train each expert $M_i$ on its own data $D_i$ [52]. We found that this method causes the experts to diverge too much from one another and from the original seed model, which makes merging after isolated training difficult.

To prevent such divergence, we train experts independently while teaching them to coordinate (Figure 1). We use $M_{\text{pub}}$ as an anchor that teaches experts to coordinate with $M_{\text{pub}}$ and, by extension, with each other. Specifically, during training, for dataset $D_i$, we construct a MoE model with two expert modules—both initialized from the same FFNs from $M_{\text{pub}}$. During training, we freeze $M_{\text{pub}}$ expert and the shared attention layer, while the other expert ($M_i$) is trained on $D_i$. As each data owner updates only their own FFNs while keeping all other parameters (those inherited from $M_{\text{pub}}$ such as attention layer) frozen, the learned FFNs are designed to naturally coordinate with each other later during merging at inference time. Importantly, with this approach, a router is learned so that each expert can be integrated into a MoE architecture without additional training (details in §3.3.2).

### 3.3.2 Domain-Informed Router

The router plays a critical role in MoE: the router function $r$ maps an input vector $\mathbf{x}$ to a distribution over expert modules, including the public model as one of the experts:
$$r(\mathbf{x}) = \mathbf{W}_r \mathbf{x}, \quad \mathbf{W}_r \in \mathbb{R}^{(n+1) \times h}$$
In typical MoEs, $\mathbf{W}_r$ is trained end-to-end alongside all expert modules, using access to the full training dataset. Instead, we decompose $\mathbf{W}_r$ into individual expert-specific router embeddings, where each row $\mathbf{r}_i$ represents the router embedding for expert $M_i$, learned only from $D_i$:

$$\mathbf{W}_r = \begin{bmatrix} -\!\!\!- & \mathbf{r}_{\text{pub}} & -\!\!\!- \\ -\!\!\!- & \mathbf{r}_1 & -\!\!\!- \\ & \vdots & \\ -\!\!\!- & \mathbf{r}_n & -\!\!\!- \end{bmatrix}, \text{ where } \mathbf{r}_i = \frac{1}{|S_i|} \sum_{d_k \in S_i} \mathbf{E}(d_k) \in \mathbb{R}^h, S_i \subset D_i.$$

These router embeddings can be initialized by averaging domain-specific embeddings of samples from each $D_i$, obtained by encoding subsets of data using an off-the-shelf embedder $\mathbf{E}$ [67] that maps a document into an $h$-dimensional vector. This method is motivated by prior model merging work that leverages domain embeddings for routing [68, 42, 69, 70, 71, 72, 73].

During coordinated training of experts (§3.3.1), we learn the router embeddings in pairs: $[\mathbf{r}_{\text{pub}}, \mathbf{r}_i]$ The public embedding $\mathbf{r}_{\text{pub}}$ remains frozen across all experts, while $\mathbf{r}_i$ is finetuned separately alongside the parameters of $M_i$. At inference time, merging the expert embeddings into the complete router matrix $\mathbf{W}_r$ directly integrates all expert modules into one unified MoE. Furthermore, experts can be flexibly added or removed by simply adding or removing their corresponding router embedding.

**Adding a Bias Term** Unlike standard router learning that is learned among all experts jointly, coordinated training of experts only learns pairwise routing decisions between one expert and the public model. This means the model never directly compares experts $M_1$ and $M_2$ during training, potentially limiting generalization during inference. To alleviate this issue, we add a negative bias term $b_i$ for each independent trained expert $\{M_1, M_2, \dots, M_n\}$. We select expert $M_i$ when:
$$\mathbf{r}_i \cdot \mathbf{x} + b_i > \mathbf{r}_{\text{pub}} \cdot \mathbf{x} \quad \forall i \in \{1, 2, ..., n\}$$
Otherwise default to $M_{\text{pub}}$. This helps the later merging process, where each expert competes not just with the public model but with all other experts. Further details and justifications are provided in §D.

### 3.3.3 Optional Router Training on Proxy Data

With our proposed model design, expert modules can be merged without any additional training. However, if data owners are willing to identify proxy samples within the public dataset $M_{\text{pub}}$ that *resemble* their closed data, we can optionally perform a lightweight router tuning step after merging, using only public data from $D_{\text{pub}}$. Specifically, we assume each data owner selects a small proxy set $\hat{D}_i \subseteq D_{\text{pub}}$, where $|\hat{D}_i| \ll 0.01 \times |D_i|$, chosen to approximate the distribution of their closed dataset $D_i$. While $\hat{D}_i$ is too small to train expert modules, it still provides useful signals for improving router quality. To construct $\hat{D}_i$, we train a binary classifier to distinguish $D_i$ from $D_{\text{pub}}$ and select public samples with the highest predicted likelihood of belonging to $D_i$. After merging, we tune the router embeddings $\mathbf{r}_1, \cdots, \mathbf{r}_n, \mathbf{r}_{\text{pub}}$ on the combined set $\hat{D}_1, \cdots, \hat{D}_n$, and $D_{\text{pub}}$, sampled uniformly.

# 4 Experimental Setup

## 4.1 Training Data: FLEXMIX

Our corpus comprises a single *Public Mix* and seven *closed sets*, either real or simulated, which are designed to be disjoint from each other. Figure 5 in §B provides the statistics.

- **Public Mix** represents general web text based on Common Crawl (CC) [1]. Specifically, we took the Baseline version of DCLM [74], excluding news and creative writing content (described below). This represents a public dataset that can be used without restrictions.
- **News** includes news content from DCLM-Baseline, obtained by applying the classifier from [75] and selecting documents classified as News Articles. While included in CC when downloaded, many of the original sources are subject to closed access [20].
- **Creative Writing** includes creative content from DCLM-Baseline, obtained by applying the classifier from [75] and selecting documents classified as Creative Writing.
- **Code** includes code repositories from Starcoder [76, 77] with additional quality filtering as in [78].
- **Academic** includes open-access academic papers obtained from [79]; these are papers from [80, 81] but re-processed using olmOCR [79] for cleaner plain text.
- **Educational Text** includes educational text from digitized PDFs, converted to plain text using olmOCR [79].
- **Math** includes math-relevant content, including web pages about or using math and math problem sets, obtained by combining Dolmino Math Mix [78] and FineMath4+ [82].
- **Reddit** contains posts and comments originally sourced and released by Dolma [83], further filtered and processed to improve quality (details in Appendix B). As of this writing, this Reddit data is no longer unrestrictedly downloadable due to Reddit's 2023 policy change.[2]

These seven sets are designed to represent datasets with at least one of the following characteristics: (1) historically closed and not publicly available; (2) previously publicly available but now closed; or (3) domains with scarce high-quality public data.

## 4.2 Evaluation

We evaluate our models and baselines on a large and diverse collection of well-established benchmarks, consisting of 31 tasks across 10 categories, broadly grouped into (1) general-purpose LM benchmarks and (2) domain-specific evaluations. More details are provided in §C.

**General-purpose Evaluation** We report results on (1) **MC9**, nine multiple-choice datasets including ARC-Easy [84], ARC-Challenge [84], BoolQ [85], CSQA [86], HellaSwag [87], OpenBookQA [88], PIQA [89], SocialIQa [90], and WinoGrande [91], (2) **GEN5**, five generative tasks including CoQA [92], SQuAD [93], Natural Questions [94], TriviaQA [95], and DROP [96], as well as (3) **MMLU** [97], (4) **MMLU-Pro** [98], (5) **AGIEval** [99] consisting of 20 tasks from college admission tasks, and (6) **BBH** [100] consisting of 23 challenging BIG-Bench tasks.

**Domain-specific Evaluation** While general-purpose evaluation benchmarks already include some math assessment, we further evaluate math ability using (6) **Math2**, which encompasses two specialized math benchmarks: GSM8K [101] and MATH [102]. To evaluate coding capabilities, we use (7) **Code4**, 4 coding benchmarks including MBPP [103], MBPPPLUS [104], HUMANEVAL [105], and HUMANEVALPLUS [104]. To measure scientific literature understanding, we report on (8) **SciRIFF5**: comprising 5 subtasks from SciRIFF [106]. Finally, we include (9) **NewsG**: news generation and (10) **PoemG**: poem generation tasks, both evaluated using an LM judge.

## 4.3 Baselines

We compare our method against several baselines, either taken directly from prior work or minimally adapted to our problem setting. All baselines, except for 'Unrestricted training,' train a set of dense

---

[1] https://commoncrawl.org

[2] Reddit's 2023 policy change restricts third-party access and use of its data, including for language model development; see nytimes.com/2023/04/18/technology/reddit-ai-openai-google.html.

models independently by continuing pretraining from the public model $M_{\text{pub}}$ on each simulated closed set, without architectural changes, and merge them using model merging techniques.

**Prompt-based Routing**   We use an LM-based domain classifier via prompting to route each query to the most suitable model, which is then used exclusively. We use Llama-3.1-8B-Instruct [107] and OLMo-2-1124-7B-Instruct [78] as classifiers. More details are in §A.1.

**Model Soup**   We apply both average and weighted parameter averaging across all models, following [12]. The weights are derived by applying a softmax over the log-likelihoods of each model on the test example input.

**Branch-Train-Merge (BTM)**   We follow BTM [11], which ensembles models by computing a weighted average of their output probabilities. Weights are obtained via a softmax over the log-likelihoods of the test example input of each model. As in the original BTM, ensembling can be restricted to the top-$k$ models by zeroing the weights of all other models and renormalizing. See §A.1 for full details.

**BTX**   We follow BTX [55], which upcycles an MoE from independently trained dense models. It copies the dense model parameters to the corresponding experts in MoE while averaging non-expert parameters such as attention layers for merging. The original BTX requires training all model parameters on combined datasets after merging. To approximate it as closely as possible while adhering to our setting, we perform this post-merge training on the public set only.

**Unrestricted MoE**   To assess how closely our method approaches the benefits of full data access while preserving data separation, we construct an upper-bound reference model: a sparse MoE initialized from the public-only dense model and trained on the combined dataset, including all closed sets and Public Mix. As MoE training incurs roughly $2\times$ the FLOPs of our approach for the same data size, we report both compute-controlled ($1\times$ FLOPs, $0.5\times$ data) and data-controlled ($2\times$ FLOPs, $1\times$ data) comparisons.

## 4.4   Training Setup

For the public model $M_{\text{pub}}$, we use a dense model with 7 billion parameters following the OLMo 2 architecture [78]. This model contains 32 layers with hidden dimension 4,096 and is trained on our public mix for 1 trillion tokens. Following [78], we use a learning rate of 0.0009 and the AdamW optimizer with parameters $\beta_1 = 0.9$ and $\beta_2 = 0.95$ and a cosine learning rate scheduler. The public model is pretrained using 512 H100 GPUs with a global batch size of 4 million tokens for three days.

Each data owner then takes this checkpoint and performs continued-pretraining for 50 billion tokens on their own data (totaling 400B tokens across all experts). For the optional router training, we use 5 billion tokens in total. The final FLEXOLMO, trained on 8 sets, has 37 billion total parameters with 20 billion active (4 active experts out of 8). More details can be found in §A.2.

## 5   Results and Analysis

We conduct ablation studies and compare against a comprehensive set of baselines at a small scale with four experts—Public mix, math, educational text, and code (Table 1). We then evaluate our final model on the full setup including the Public mix and all seven simulated closed sets (Table 2). Finally, we present an in-depth analysis to illustrate the behavior and effectiveness of FLEXOLMO.

## 5.1   Main Results

**Individual experts excel at their specialized tasks**   As shown in Table 1, experts trained on each domain-specific set demonstrate strong performance in their specialized domains: the Math expert achieves the highest scores on math tasks, while the Code expert performs the best on coding benchmarks. However, these experts exhibit considerable performance degradation when evaluated on tasks outside their domains. Notably, the Code expert performs poorly on general benchmarks.

**FLEXOLMO outperforms individual experts**   FLEXOLMO outperforms individual experts in most cases. It improves upon the model trained solely on public data, achieving an average 41% relative gain. Largest improvements appear on benchmarks where closed data significantly boosts

Table 1: Evaluation of FLEXOLMO trained on *four* sets (public mix, math, educational text and code), tested on 24 tasks with 100 samples per subtask.

| | MC9 | GEN5 | MMLU | MMLU Pro | AGI Eval | BBH | Math2 | Code4 | Avg. |
|---|---|---|---|---|---|---|---|---|---|
| Prev. Public model | 68.4 | 58.8 | 57.0 | 27.1 | 39.0 | 35.6 | 8.1 | 1.0 | 36.9 |
| *Individual experts* | | | | | | | | | |
| Math | 63.8 | 46.3 | 51.1 | 24.0 | 40.7 | 45.4 | 50.4 | 18.1 | 42.5 |
| Code | 38.7 | 41.4 | 30.0 | 14.6 | 29.0 | 38.2 | 6.0 | **22.4** | 27.5 |
| Educational Text | 63.0 | 52.8 | 57.7 | 26.8 | 39.6 | 40.0 | 13.1 | 4.3 | 37.2 |
| *Prior model merging work* | | | | | | | | | |
| Model soup (average) | 70.6 | 53.8 | 54.7 | 28.4 | 41.4 | 42.4 | 17.5 | 8.2 | 39.6 |
| Model soup (weighted) | 69.6 | 56.3 | 58.6 | 30.5 | 45.4 | 43.5 | 18.5 | 14.8 | 42.2 |
| BTM | 69.0 | 58.5 | 59.6 | 29.0 | 43.6 | 43.6 | 21.2 | 22.3 | 43.4 |
| Prompt-based routing (router: OLMo) | 59.9 | 48.8 | 50.0 | 25.4 | 38.7 | 41.3 | 41.5 | 20.7 | 40.8 |
| Prompt-based routing (router: Llama) | 64.2 | 53.4 | 57.7 | 26.4 | 39.9 | 39.8 | 21.5 | 17.3 | 40.0 |
| BTX | 69.6 | 57.9 | 56.2 | 28.5 | 43.1 | 41.3 | 16.8 | 6.4 | 40.0 |
| *Ours* | | | | | | | | | |
| FLEXOLMO (no optional router training) | **71.1** | 58.6 | 58.1 | 28.4 | 44.8 | 43.4 | **51.5** | 18.2 | 46.7 |
|   - no bias | 67.9 | 55.6 | 57.5 | 28.6 | 43.9 | 45.5 | 50.0 | 17.6 | 45.8 |
|   - no domain embedding init, no bias | 70.0 | 55.4 | 56.1 | 25.9 | 41.1 | 44.9 | 44.9 | 16.6 | 44.4 |
|   - no training to coordinate | 64.4 | 51.5 | 55.5 | 24.7 | 43.1 | 41.2 | 19.3 | 10.3 | 38.8 |
| FLEXOLMO | 71.0 | **59.8** | **59.9** | **30.8** | **45.8** | **47.1** | 50.7 | 17.3 | **47.8** |
| *Reference model (upperbound)* | | | | | | | | | |
| Unrestricted MoE (1× FLOPs, 0.5× Data) | 68.0 | 53.8 | 57.8 | 28.9 | 41.5 | 48.6 | 49.4 | 22.2 | 46.3 |
| Unrestricted MoE (2× FLOPs, 1× Data) | 73.3 | 60.2 | 63.1 | 32.5 | 48.1 | 54.4 | 53.4 | 27.0 | 51.5 |

Table 2: Evaluation of FLEXOLMO trained on *eight* sets (public mix and seven simulated closed sets) on 31 tasks across 10 categories, tested with 1,000 samples per subtask. "no RT" indicates no optional router training on proxy data (§3.3.3).

| | MC9 | GEN5 | MMLU | MMLU Pro | AGIEval | BBH | Math2 | NewsG | PoemG | SciRIFF5 | Code4 | Avg. |
|---|---|---|---|---|---|---|---|---|---|---|---|---|
| Prev. Public model | 68.7 | 58.8 | 55.9 | 26.2 | 39.9 | 35.7 | 8.2 | 76.0 | 47.8 | 48.1 | 1.1 | 42.4 |
| *Individual experts* | | | | | | | | | | | | |
| Math | 62.5 | 44.3 | 50.6 | 24.1 | 42.0 | 45.6 | **53.1** | 42.6 | 28.0 | 50.7 | 15.8 | 41.8 |
| Code | 40.5 | 39.4 | 29.5 | 14.5 | 27.4 | 38.1 | 6.0 | 45.1 | 28.2 | 48.0 | 21.0 | 30.7 |
| Educational Text | 64.3 | 52.1 | 56.5 | 27.0 | 39.7 | 40.3 | 13.6 | 57.6 | 51.8 | 51.7 | 3.0 | 41.6 |
| News | 46.5 | 48.6 | 36.4 | 15.2 | 25.7 | 30.9 | 2.5 | 77.7 | 26.9 | 47.0 | 0.0 | 32.5 |
| Creative Writing | 42.7 | 43.9 | 31.5 | 11.6 | 23.3 | 27.6 | 1.7 | 56.9 | **67.5** | 42.4 | 0.0 | 31.7 |
| Academic | 41.0 | 45.2 | 33.8 | 14.8 | 24.1 | 32.4 | 6.5 | 51.8 | 23.0 | 52.0 | 0.0 | 29.5 |
| Reddit | 64.7 | 36.5 | 56.1 | 25.5 | 35.5 | 19.7 | 2.5 | 54.1 | 8.6 | 32.7 | 1.7 | 30.7 |
| *Combined model* | | | | | | | | | | | | |
| BTM (top-2) | 68.7 | 57.7 | 59.4 | 28.3 | 43.2 | 44.3 | 23.1 | 73.6 | 54.4 | 46.3 | **24.0** | 47.6 |
| FLEXOLMO (no RT) | 69.2 | 53.2 | 58.8 | **34.0** | 43.4 | 42.1 | 52.1 | 78.2 | 60.1 | **54.4** | 18.6 | 51.3 |
| FLEXOLMO | **70.8** | **59.8** | **60.4** | 30.9 | **45.1** | **46.4** | 48.5 | **80.7** | 62.2 | 54.3 | 17.2 | **52.4** |

individual expert performance, e.g., $35.6 \rightarrow 47.1$ on BBH, $8.1 \rightarrow 50.7$ on math, and $1.0 \rightarrow 17.3$ on coding. Notably, FLEXOLMO even matches or exceeds the performance of specialized experts on their respective tasks (e.g., on BBH and Math2).

**FLEXOLMO achieves more effective merging than baselines** We also compare FLEXOLMO with baseline merging methods (§4.3). All baselines outperform the model trained on Public Mix only. However, their performance is inconsistent: model soup and BTX are generally weak,[3] while prompt-based routing is highly unstable: it performs well when the classifier selects the correct expert, but degrades sharply when it does not. Among the baselines, BTM yields the best performance. Nonetheless, FLEXOLMO outperforms all prior model merging methods, beating the best baseline BTM by 10.1% relative on average. We attribute this to the MoE-based design of our model, which selectively activates different experts per layer, effectively combining the complementary strengths of each specialized model (see further analysis in §5.2).

---

[3]This is likely because training on disjoint datasets causes experts to diverge from each other and from the seed model, making model soup limited, and training BTX on the public data only is not optimal.

Table 3: **Impact of embedding initialization methods on model performance.** The GRIT embedder [67] consistently outperforms public model embeddings across most benchmarks.

|  | Embed Init. | MC9 | GEN5 | MMLU | MMLU Pro | AGI Eval | BBH | Math2 | Code4 | Avg. |
|---|---|---|---|---|---|---|---|---|---|---|
| FLEXOLMO | Public | 70.5 | 55.5 | **58.3** | 26.5 | 40.2 | 40.1 | 48.4 | 8.7 | 43.5 |
| FLEXOLMO | GRIT | **71.1** | **58.6** | 58.1 | **28.4** | **44.8** | **43.4** | **51.5** | **18.2** | **46.7** |

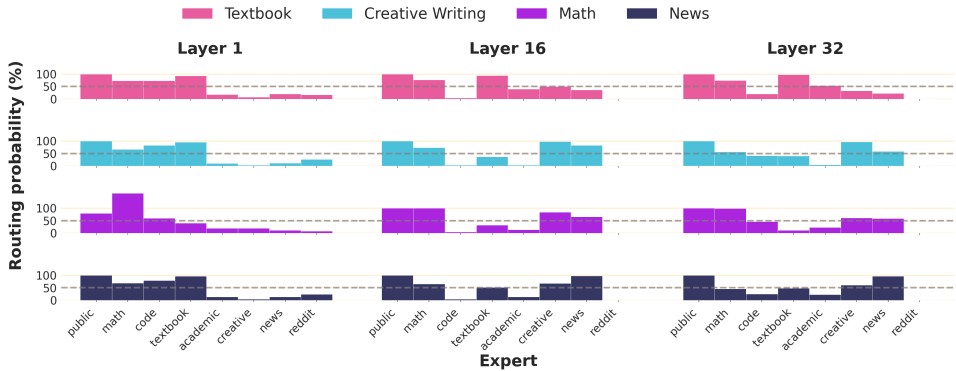

Figure 2: **Routing pattern analysis**. We visualize how text from different domains activate experts (four experts activated). The horizontal gray lines indicate uniform routing.

**Comparison to the unrestricted MoE** Compared to the unrestricted MoE trained without considering data restrictions (§4.3), FLEXOLMO outperforms the FLOP-controlled setting ($1\times$ FLOPs, $0.5\times$ Data). It slightly underperforms the data-controlled model ($2\times$ FLOPs, $1\times$ Data). This indicates that FLEXOLMO enables training without direct access to the data (requiring only model sharing) and flexible opt-in and opt-out functions *while* retaining strong performance.

**Ablations on FLEXOLMO** We further evaluate FLEXOLMO by removing different components introduced in §3.3: learning to coordinate, router initialization, and the bias term. Our results show that each component plays an important role, with the removal of any one leading to performance drop. In particular, we observe that randomly initializing router embeddings leads to the final learned router embeddings being very similar to each other, making the later merging of multiple experts harder. Furthermore, we confirm that FLEXOLMO benefits from additional router training (§3.3.3) and using external embedders for router initialization, compared to using the public model's hidden states as the initialization (Table 3).

**Final FLEXOLMO in the full setup** Finally, we evaluate FLEXOLMO in the full eight-expert setup and compare it against the public-only model, individual experts trained on closed datasets, and BTM (top-2), the strongest baseline from Table 1. This was done by simply *adding four additional experts*, benefiting from FLEXOLMO's flexibility in easily adding new datasets. Consistent with earlier findings, FLEXOLMO outperforms all individual models (the public baseline and individual experts), demonstrating the synergistic effect of combining independently trained modules. Compared to the strongest baseline BTM, it achieves a 10% relative improvement on average (Table 2). FLEXOLMO excels on benchmarks where specialized experts perform well (BBH, Math2, NewsG, PoemG, SciRIFF5, Code4), matching or surpassing the experts, and also shows strong results on tasks where no single dataset suffices (e.g., MC9, Gen5, MMLU, MMLU Pro, AGI Eval).

## 5.2 Model Behavior Analysis

**Routing patterns** Figure 2 visualizes the router's token distribution across experts for various domain inputs. The router tends to activate the corresponding domain expert (e.g., math inputs activate the math expert), demonstrating its ability to identify the most relevant module. We also observe frequent activation of the public expert, likely due to our coordinated training strategy, where each expert is designed to complement the public expert. Also, different combinations of experts are activated at different layers. This highlights the model's layer-specific specialization and its greater expressivity than approaches that route inputs to a single expert (e.g., prompt-based routing).

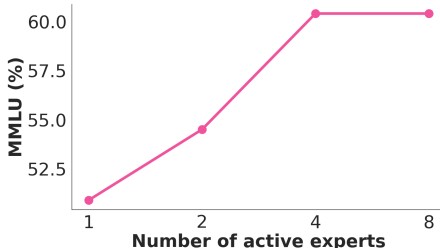

Figure 3: **Effect of active expert count**

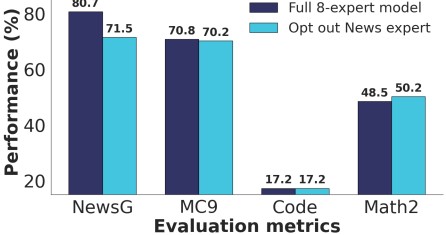

Figure 4: **Opting out of news data**.

**Number of active experts**  We further analyze how the number of active experts affects downstream task performance. As shown in Figure 3, performance consistently improves as more experts are activated, up to four experts, after which it plateaus. This suggests that the final model can operate efficiently as a sparse model by activating only four experts per input during inference.

**Data opt-out**  FLEXOLMO offers a straightforward mechanism for opting out of specific datasets by removing the corresponding expert module at inference time. In Figure 4, we evaluate the model's performance after excluding the news expert. As expected, performance drops on in-domain tasks such as news generation. However, on unrelated tasks, performance remains largely unaffected.

### 5.3  Data Extraction Analysis

FLEXOLMO enables data owners to contribute to the model without sharing their data by instead sharing model weights trained locally on their data. A natural and important concern is whether their data can be extracted from these shared weights [108, 109, 110]. This risk is particularly relevant when training data includes private or confidential information.

We empirically assess this risk by implementing training data extraction following prior work [108]. Specifically, we sample 10,000 documents from the math data.[4] From each document, we extract a 32-token prefix and use it to generate 256-token continuation using top-$k$ sampling ($k = 50$), top-$p$ sampling ($p = 0.95$), and a temperature of $1.0$. We sample 10 times per prefix, and if any of the generated outputs achieves a normalized Levenshtein similarity of 0.9 or higher with the original document, we consider that document to be extracted. To validate our implementation, we apply it to a model overfitted on the dataset (trained for 100 epochs), and observe a 60% extraction rate.

Our results are as follows: (1) A public model that has not seen any math data yields an extraction rate of 0.1%. (2) A dense model trained on the math dataset (i.e., math expert) yields 1.6%. (3) FLEXOLMO with the math expert included yields 0.7%.

These results lead to the following conclusions. First, in practice, it is difficult to extract a substantial portion of the training data, which is in line with previous findings [108]. However, if a model includes any weights trained on the data, nonzero (though small) fraction of the data may be extractable. If data owners are comfortable with this minimal leakage, as long as a meaningful fraction of the data remains not extractable, we believe that FLEXOLMO, in its current form, is a viable solution. If the owners' data includes any private or sensitive information, we recommend training experts using differentially private (DP) learning methods before contributing them to the model, which provides formal privacy guarantees. Applying DP is orthogonal to our architecture, and different data owners can make independent decisions, providing flexibility without compromising the overall design.

## 6  Conclusion

We introduce FLEXOLMO, a new class of LMs that solves real-world data constraint challenges with (1) *modular, distributed training*, where different model parameters are independently trained on disjoint and locally maintained datasets, and (2) *data-flexible inference*, where data can be selectively included at inference-time, with guarantees. We show that FLEXOLMO significantly outperforms competitive baselines, while providing the benefits of distributed training and flexible inference.

---

[4]We chose the math data because it is the smallest among our simulated closed sets and the math expert is trained for three epochs (instead of one), making it more susceptible to extraction. Therefore, the extraction rates with the math data likely represent an upper bound.

## Acknowledgements

We thank Preston Jiang, Colin Raffel, Percy Liang, Matei Zaharia, Peter Henderson, David Q. Sun, Kevin Kuo, Virginia Smith, and Ai2 members for valuable discussion and feedback.

SM was supported in part by a grant from DARPA to the Simons Institute for the Theory of Computing. PWK was supported by the Singapore National Research Foundation and the National AI Group in the Singapore Ministry of Digital Development and Information under the AI Visiting Professorship Programme (award number AIVP-2024-001), and by the AI2050 program at Schmidt Sciences.

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

# A  Model Details

## A.1  Baseline Details

**Prompt-based routing**  We implemented a domain-specific classifier where each input query is categorized into exactly one of $n$ domains using the prompts shown below. Once classified, the query is routed exclusively to the corresponding expert model for processing. In particular, the following prompt was used for our primary experimental setup:

```
"""Classify the given text into one of the following domains:
- 0: Math
- 1: Code
- 2: General science and educational content
- 3: News
- 4: Creative writing and literature
- 5: Reddit
- 6: Research and academic content
- 7: Other

Provide the number between 0 and 7."""
```

For ablations with four datasets instead of eight, we used the following prompting.

```
"""Classify the given text into one of the following domains:
- 0: Math
- 1: Code
- 2: General science and academic content
- 3: Other

Provide the number between 0 and 3."""
```

**Branch–Train–Merge (BTM)**  BTM [11] first assigns a weight to each expert and then combines their predictions through ensembling.

Formally, for a test instance $x$ and the expert $i$ we compute the negative-log-likelihood loss $\mathcal{L}_i(x)$ and convert it to a weight with a temperature-controlled softmax:

$$w_i(x) = \frac{\exp(-\mathcal{L}_i(x)/\tau)}{\sum_{j=1}^{M} \exp(-\mathcal{L}_j(x)/\tau)},$$

where $\tau > 0$ sharpens ($\tau < 1$) or flattens ($\tau > 1$) the distribution over the $M$ experts. We optionally retain only the $k$ largest-weight experts $S_k(x)$ ($k < M$) and renormalize:

$$\tilde{w}_i(x) = \frac{w_i(x)\,\mathbf{1}[\,i \in S_k(x)\,]}{\sum_{j \in S_k(x)} w_j(x)}.$$

At inference time, let $\mathbf{z}_{i,t} \in \mathbb{R}^{|V|}$ be the logit vector produced by the expert $i$ at generation step $t$. We combine logits token-by-token:

$$z_t(v) = \sum_{i \in S_k(x)} \tilde{w}_i(x)\,z_{i,t}(v), \qquad v \in V.$$

## A.2  Training Details

In FLEXOLMO, to initialize router embeddings with domain embeddings, we sample 1,000 documents from each data source, process them through `GritLM/GritLM-7B` [13] to obtain document embeddings, and then average these embeddings.

To obtain proxy data $\hat{D}_i$, we train a binary classifier to distinguish $D_i$ from $D_{\text{pub}}$ and select public samples with the highest predicted likelihood of belonging to $D_i$. Specifically, we finetune `Snowflake/snowflake-arctic-embed-xs` [111], which contains 22M parameters, using a learning rate of $3 \times 10^{-6}$. The classifier is trained on a balanced dataset of 500,000 samples (250,000 documents from each source - public and private). The classifier quickly achieved an accuracy above 95% across all datasets considered.

# B Data Details

Table 5 presents the statistics of our training data described in Section 4.1.

## B.1 Reddit Data Processing

The construction of this dataset involved three major phases.

**1. Reddit data filtering** A dataset of submission/comment pairs was derived from the PushShift Reddit dataset [112] (bulk dump as of March 2023) – the same dump used for Dolma Reddit (`https://huggingface.co/datasets/allenai/dolma`).

To derive our initial dataset, we extracted each submission and concatenated it with its top-scoring, top-level comment. We then performed further rule-based filtering with the following constraints:

- Filter out deleted/removed content.
- Filter out content marked as over_18.
- Filter out all posts from a list of 26,123 banned or NSFW subreddits.
- Filter out posts from likely bot authors (drawn from https://botrank.pastimes.eu/ as of Sept 2024).
- Filter out posts containing non-text media.
- Perform document-level text deduplication via Bloom filter.

Figure 5: Statistics of our data mix (descriptions in §4.1).

| Name | # Tokens (B) |
|------|-------------|
| Public Mix | $2.37 \times 10^3$ |
| News | 158.0 |
| Creative Writing | 201.9 |
| Math | 20.3 |
| StarCoder | 83.0 |
| Academic | 58.6 |
| Educational text | 102.2 |
| Reddit | 9.9 |

**2. Retrieval-based subreddit selection** Dense retrieval was then used to identify academically-relevant subreddits for further filtering. We adapted search queries from MMLU test questions, and performed dense retrieval with these queries on the filtered Reddit data from Step #2, retaining the top 5 hits for each query. Based on these retrieved outputs, we selected 151 subreddits meeting the following criteria:

- Subreddit has >= 20 *unique* retrieved items for queries within a given MMLU category; OR
- Subreddit has >=100 retrieved items for queries across all MMLU categories.

We then filtered the dataset from Step #1 to retain only documents from subreddits on this list of 151 subreddits.

**3. Format rewriting** Finally, the data from Step #2 was input to a synthetic rewriting pipeline to generate academic QA items with coverage of diverse question formats. We defined 7 categories of question format inspired by variation observed in MMLU, and used these to construct prompts for QA text generation. The format categories are as follows:

1. open-ended
2. statement completion
3. fill-in-the-blank
4. statement truth verification
5. which-of-following-has-property-X
6. which-of-following-is-true
7. in-question options

For each format category we constructed a prompt for generating questions of that category given an input text. Below is an example prompt, for the "in-question-options" category. Prompts for other categories differ in 1) the content of the "For format ..." paragraph and 2) the in-context examples (1-3 examples per prompt).

*I will ask you to convert a text into multiple-choice questions. Here is the text:*

*"{text}"*

*Instructions: Convert the information in the text into academic multiple choice questions. ONLY include questions that are academic. DONOT reference the text in the question.*

*For format, use questions that provide options within the question and give choices for which options are true. Examples:*

*Dogs have which of the following properties?*

*I. They are mammals*
*II. They have five legs.*
*III. They have a tail.*

*A. I only*
*B. II only*
*C. III only*
*D. I and III*

*Answer: D*

*%%%*

*Which of the following are cities in the US?*

*I. Paris*
*II. Athens*
*III. Chicago*

*A. I only*
*B. II only*
*C. III only*
*D. I, II and III*

*Answer: C*

*Separate ALL questions with "\n%%%\n".*

For generating our rewritten QA data, we prompted GPT-4o mini (Jan 2025 version). We iterated over the submission/comment pairs in the data from Step #2, and for each of these texts we sampled a format category and prompted the GPT-4o mini to generate QA pairs for that text and format category. For longer input texts, format categories were resampled and prompted for again, a number of times proportional to the length of the text.

Finally, GPT-4o mini outputs were parsed into separate QA items based on the "%%%" separator, and 50% of items were prepended with the prefix "Question: ".

We validated these rewritten data in experiments with OLMo 7B [113] models trained to 2T tokens, carrying out continued pretraining on a 50-50 mix of DCLM and Reddit data while annealing the learning rate to zero, a strategy used in [114, 78, 107]. We run this continued pretraining with two versions of Reddit data: the filtered data from Step #2, and the rewritten data from Step #3. We find that the rewriting improves over the non-rewritten data in both MC9 and MMLU: **MC9 improves from 0.74 to 0.76** and **MMLU improves from 0.62 to 0.66**.

Table 4: **Evaluation benchmarks.** . Benchmarks are divided into general-purpose and domain-specific categories. Original dataset citations are listed at right. †SCIRIFF spans several subtasks (BioASQ factoid, general, and yes/no questions, COVID DeepSet QA, and PubMedQA) evaluated with different metrics. MATH† includes seven subsets: algebra, counting and probability, geometry, intermediate algebra, number theory, prealgebra, and precalculus.

| General Benchmarks | | | Domain-specific Benchmarks | | |
|---|---|---|---|---|---|
| **Benchmark** | **Metric** | **Citation** | **Benchmark** | **Metric** | **Citation** |
| ARC-EASY | $Acc_{raw}$ | [84] | HUMANEVAL | Pass@1 | [105] |
| ARC-CHALLENGE | $Acc_{raw}$ | [84] | HUMANEVALPLUS | Pass@1 | [104] |
| BOOLQ | $Acc_{raw}$ | [85] | MBPP | Pass@1 | [103] |
| CSQA | $Acc_{raw}$ | [86] | MBPPPLUS | Pass@1 | [104] |
| HELLASWAG | $Acc_{raw}$ | [87] | SCIRIFF† | — | [106] |
| OPENBOOKQA | $Acc_{raw}$ | [88] | MATH† | Exact Match | [102] |
| PIQA | $Acc_{raw}$ | [89] | NEWS GENERATION | LM-as-judge | - |
| SOCIAL IQA | $Acc_{raw}$ | [90] | POEM GENERATION | LM-as-judge | - |
| WINOGRANDE | $Acc_{raw}$ | [91] | GSM8K | Exact Match | [101] |
| MMLU | $Acc_{raw}$ | [97] | | | |
| MMLU-PRO | $Acc_{raw}$ | [98] | | | |
| AGIEVAL (English) | $Acc_{raw}$ | [99] | | | |
| BIG-BENCH HARD (BBH) | Exact Match | [100] | | | |
| CoQA | F1 | [92] | | | |
| DROP | F1 | [96] | | | |
| NATURAL QUESTIONS (NQ) | F1 | [94] | | | |
| SQUAD | F1 | [93] | | | |
| TRIVIAQA | F1 | [95] | | | |
| NARRATIVEQA | F1 | [115] | | | |

*Aggregate scores (averages of individual benchmarks)*
GEN5: CoQA, SQUAD, NATURAL QUESTIONS, TRIVIAQA, DROP
MC9: ARC-EASY, ARC-CHALLENGE, BOOLQ, CSQA, HELLASWAG, OPENBOOKQA, PIQA, SOCIAL IQA, WINOGRANDE
CODE4: MBPP, MBPPPLUS, HUMANEVAL, HUMANEVALPLUS

## C  Evaluation Details

All evaluations are done using the OLMES evaluation standard introduced by [116], following key metrics from the OLMO 2 framework [78]. Table 4 presents a detailed breakdown of our evaluation datasets and metrics. General description is provided in §4.2; here, we provide more details.

**SciRiFF**  We select five subtasks that do not require structured prediction: `BioASQ-Factoid`, `BioASQ-Yes/No`, `BioASQ-General`, `PubMedQA`, and `COVID-QA`. , with our tables reporting the average performance across all five tasks.

**MATH**  We assess solution correctness through exact matching with ground truth answers, following the methodology established in [117].

**News Generation**  We use the `muse-bench/MUSE-News` dataset from [118], selecting articles containing between 64 and 128 tokens. Models are prompted to continue an article given a prefix of the first 32 tokens, using a sampling temperature of 0.8. A `Llama-3.3-70B-Instruct` model serves as the judge, evaluating each completion based on journalistic quality (2 points), topical coherence (2 points), and clarity/fluency (1 point), for a total score between 0 and 5, which is then normalized to a 0–100 scale. To reduce variance, we generate five completions per prompt and report the average score.

**Poem Generation**  We employ the `merve/poetry` dataset[5], filtering poems to those containing 64–128 tokens. From a total of 176 poems (147 Renaissance, 29 Modern), we reserve five poems (three Renaissance, two Modern) for few-shot examples and use 100 for testing. Models continue each poem from its first four lines, following the same prompting and evaluation settings as in NEWSGEN. A genre-aware `Llama-3.3-70B-Instruct` judge evaluates each completion based on poetic craftsmanship (2 points), thematic coherence (2 points), and clarity/fluency (1 point), with

---

[5]https://www.kaggle.com/datasets/ishnoor/poetry-analysis-with-machine-learning/data

scores normalized to a 0–100 scale. As with news generation, five completions are generated per prompt, and we report the average score across all instances.

# D  Methodology Intuition

## D.1  Problem Setup

Given an input data $\mathbf{x} \in \mathbb{R}^h$, the router learning can be viewed as a multi-class classification problem with $n+1$ classes: a public class $\mathcal{C}_{\text{pub}}$ and $n$ closed classes $\{\mathcal{C}_i\}_{i=1}^n$. The scoring function $s_i : \mathbb{R}^h \to \mathbb{R}$ such that $s_i(\mathbf{x}) = \mathbf{r}_i \cdot \mathbf{x}$ represents the score for class $\mathcal{C}_i$, where $\mathbf{r_i}$ is the router embedding and the higher scores indicate higher class membership likelihood.

**Training: Pairwise Binary Classification** In training experts to coordinate (§3.3.1), we learn $n$ binary classifiers $\{f_i\}_{i=1}^n$, where each $f_i : \mathbb{R}^h \to \{\mathcal{C}_{\text{pub}}, \mathcal{C}_i\}$ discriminates between $\mathcal{C}_{\text{pub}}$ and $\mathcal{C}_i$:

$$f_i(\mathbf{x}) = \begin{cases} \mathcal{C}_i & \text{if } s_i(\mathbf{x}) > s_{\text{pub}}(\mathbf{x}) \\ \mathcal{C}_{\text{pub}} & \text{otherwise.} \end{cases}$$

The decision boundary $h_i$ between $\mathcal{C}_{\text{pub}}$ and $\mathcal{C}_i$ is defined as:

$$h_i := \{\mathbf{x} \in \mathbb{R}^h : s_{\text{pub}}(\mathbf{x}) = s_i(\mathbf{x})\}.$$

**Inference: Multiclass Classification** At inference time (§ 3.3.2), these $n$ binary classifiers are combined into a unified multi-class classifier that maps from $\mathbb{R}^h$ to the complete class space $\mathcal{C}_{\text{pub}}, \mathcal{C}_1, \ldots, \mathcal{C}_n$. The final classification decision is determined by:

$$F(\mathbf{x}) = \arg \max_{i \in \{\text{pub}, 1, \cdots, n\}} s_i(\mathbf{x}).$$

The key question is: how could we make the ensemble of binary classifiers $\{f_i\}_{i=1}^n$ route inputs as close to if we had trained one unified multiclass classifier $F$ end-to-end on all data?

Our intuition is if each binary classifier $f_i$ learns the decision boundary "$\mathcal{C}_i$ vs. not $\mathcal{C}_i$" rather than just "$\mathcal{C}_i$ vs. $\mathcal{C}_{\text{pub}}$", this could make decision boundary that tightly encircles $\mathcal{C}_i$ which leads to better multi-class classification.

## D.2  Anchor Point and Negative Bias

To ensure each classifier $f_i$ learns a better decision boundary $h_i$, we propose two key techniques:

**Freezing $r_{\text{pub}}$ and $M_{\text{pub}}$ as anchor points** During training of each binary classifier $f_i$, we fix the public expert $M_{\text{pub}}$ and router embedding $\mathbf{r}_{\text{pub}}$ to ensure all classifiers share a common coordinate system by maintaining:

$$s_{\text{pub}}(\mathbf{x}) = \mathbf{r}_{\text{pub}} \cdot \mathbf{x}, \forall i \in \{1, \cdots, n\}.$$

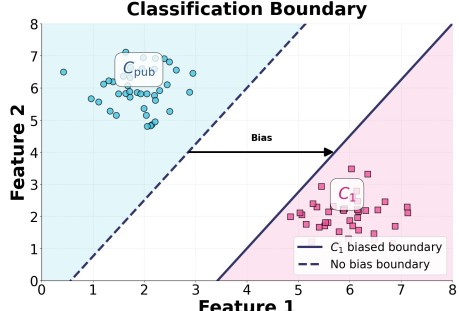

Figure 6: During training, the negative bias shifts the decision boundary. So that a more selective subset of data will be used to train the expert corresponding to $\mathcal{C}_1$.

Without this constraint, each binary classifier would optimize both $\mathbf{r}_{\text{pub}}$ and $\mathbf{r}_i$ independently, resulting in inconsistent coordinate systems where $\mathbf{r}_{\text{pub}}^{(i)} \neq \mathbf{r}_{\text{pub}}^{(j)}$ for $i \neq j$ during the training of binary classifiers $f_i$ and $f_j$. Such inconsistency would invalidate the multi-class classifier $F(\mathbf{x})$, as it would attempt to compare scores computed in incompatible embedding spaces. By maintaining a fixed reference point, we ensure all decision boundaries $h_i$ are defined relative to the same coordinate system, enabling their meaningful composition during inference without additional training.

**Adding a negative bias** During training of $f_i$, each binary classifier only needs to satisfy $\mathbf{r}_i \cdot \mathbf{x} > \mathbf{r}_{\text{pub}} \cdot \mathbf{x}$ for $\mathbf{x} \in \mathcal{C}_i$. When $\mathcal{C}_{\text{pub}}$ and $\mathcal{C}_i$ is separable, many possible $\mathbf{r}_i$ vectors (no bias boundary in Figure 6) can satisfy this constraint without precisely characterizing the specialized region of $\mathcal{C}_i$. We refine this by adding a negative bias term to each expert:

$$\mathbf{r}_i \cdot \mathbf{x} + b_i > \mathbf{r}_{\text{pub}} \cdot \mathbf{x} \quad \forall i \in \{1, 2, ..., n\}$$

As Figure 6 illustrates, the negative bias term could help move the decision boundary $h_i$ closer to $\mathcal{C}_i$'s data points. This means, during training, a more selective subset of data will be used to train the local expert. This facilitates merging, where experts compete not only with $\mathcal{C}_{\text{pub}}$ but with each other.

