# OpenReview forum: "FlexOLMo: Open Language Models for Flexible Data Use"
_NeurIPS.cc/2025/Conference — NeurIPS 2025 spotlight_

### Official Review · Reviewer_F5CK · 2025-06-23

**Clarity:** 3
**Significance:** 2
**Originality:** 3
**Rating:** 5
**Confidence:** 4

**Summary:**

This paper introduces **FlexOlmo**, a new class of language models that supports distributed training without data sharing and data-flexible inference, where parameters along with their associated data can be flexibly included or excluded with no further training.
This paper also introduces **FlexMix**, a corpus comprising seven restricted sets, either real or realistic approximations, alongside 10 publicly available datasets.

**Questions:**

**Questions**:
1. What is the motivation behind processing and filtering the original Reddit dataset? Why not directly use the same version adopted in previous studies?
2. What is the rationale for selecting these eight domains? Are there any references or prior work supporting the inclusion of domains such as "Creative Writing" or "Reddit"? Furthermore, is there any semantic relationship or overlap among these domains?
***
**Suggestions**:
1. Some sentences in the paper could be improved for clarity. For example:
> "A corpus we curate comprising seven restricted sets, either real or realistic approximations, alongside 10 publicly available datasets."

This sentence is somewhat overloaded in its current form. The phrase "we curate" may be unnecessary, as it is already implied by the context.
2. For Figure 4, I believe results on more benchmarks (not just MMLU) will be more informative.
***

**Ethical Concerns:**

["NO or VERY MINOR ethics concerns only"]

**Final Justification:**

Most of my concerns are addressed, particularly weakness 1,3,4,5.
For weakness 2, I understand there is always a risk and the proposed method draw an acceptable trade-off.

**Limitations:**

I think the paper needs to open up a separate section to discuss its limitations.

**Quality:**

2

**Strengths And Weaknesses:**

**Strengths**:
1. I like the core idea of FlexOlmo, which leverages a public dataset as a medium for coordination across domains. While conceptually straightforward, it yields meaningful performance improvements as demonstrated in the experiments.
2. The paper includes a comprehensive set of evaluation benchmarks and conducts necesary ablation study within the page limit.
3. Generally, this paper is clearly structured and easy to follow.
***
**Weaknesses**:
1. The training procedure of the expert model $M_i$ and route embedding $r_i$ are not clearly explained, particularly regarding how their parameters are jointly updated across the dataset $D_i$, in Section 3.3.1 and 3.3.2.
2. For Section 3.3.3, the distribution of private data may also be privacy-sensitive.
3. In section 2.1, the authors discuss two types of data constraints. For the first constraint on data sharing, I think decentralized training can adequatly address. However, for the second constraint on data usage, I'm not fully convinced how FlexOlmo can handle restrictions including copyright restrictions. In certain scenarios, take toxicity classification as an example, it would be helpful if the authors could provide a more detailed explanation of how FlexOlmo addresses this challenge and why it is preferable to conventional methods, such as directly fine-tuning on datasets that contain toxic content.
4. The core contribution and motivation of the constructed dataset FlexMix is unclear.  And I suggest the authors explicitly list their main contributions in the Introduction section.
5. Lack of domain-specific evaluation. Specifically, there are no corresponding evaluations for *Reddit* and *Textbook* domains.
6. From experiments, the majority of FlexOlmo's performance gains appear to come from "Math2". If the math domain is excluded, does FlexOlmo still outperform BTM and BTX?
7. I notice that the training of FlexOlmo requires substantial computational resources in Section 4.4. In practical decentralized training scenarios, however, data owners may not have access to such extensive resources for continual training. This limitation should be discussed.
***

---

> ### Author Rebuttal · Authors · 2025-07-31
>
> We thank the reviewer for their time and constructive feedback. We appreciate that you like the core idea of FlexOlmo. We address each of your specific comments below.
>
> > The training procedure of the expert model and route embedding are not clearly explained, particularly regarding how their parameters are jointly updated across the dataset, in Section 3.3.1 and 3.3.2.
>
> The training process of Flexolmo is as follows (as shown in Figure 1):
> 1. First, a public model is trained on data without any restrictions.
> 2. Then, each data owner trains their local expert alongside a frozen copy of the public model. This public model acts as a shared "anchor," ensuring that all independently trained experts learn to operate within a common functional space. During this local training stage, the public model and its public router embedding remain frozen. Only the local expert's parameters (FFN layers) and its corresponding router embedding are updated.
> 3. Finally, in the merging step, we combine the frozen public expert and all the independently trained local experts by simply concatenating their learned router embeddings into a joint table.
>
> We thank the reviewer for this feedback and will make the explanation more clear in the final version of the paper.
>
>
> > For Section 3.3.3, the distribution of private data may also be privacy-sensitive.
>
> We agree with the reviewer that the selection of public data based on the private distribution may carry the risk of indirectly revealing information about it. We clarify in the paper that this is an optional step a data owner would only undertake if they are comfortable with this risk.
>
> >  I'm not fully convinced how FlexOlmo can handle restrictions including copyright restrictions. In certain scenarios, take toxicity classification as an example, it would be helpful if the authors could provide a more detailed explanation of how FlexOlmo addresses this challenge and why it is preferable to conventional methods, such as directly fine-tuning on datasets that contain toxic content.
>
> Training a single FlexOlmo model allows for deploying different combinations of experts (and their associated data) to different users based on specific access criteria such as permissions, policies, or preferences. Here are two concrete examples:
>
> - Copyrighted Data: Suppose two newspapers contribute experts trained on their proprietary data. They could restrict access so that only users with an active subscription to their respective papers have the option to activate the associated expert, or activate both experts if the user subscribes to both, while others cannot.
> - Content Moderation: The FlexOlmo architecture allows for the on-demand deployment of specialized experts, such as one for toxicity detection. This expert can be activated specifically for sensitive applications like toxicity detection, while remaining inactive in other use cases where child-safety is important
> In summary, we train a single FlexOlmo model, but it can function as if it were trained on many different data combinations, customizing its usage to match specific deployment requirements and constraints.
>
> > Lack of domain-specific evaluation. Specifically, there are no corresponding evaluations for Reddit and Textbook domains.
>
> Thank you for this suggestion. We include many domain-specific evaluations as detailed in Section 4.2. For datasets like Textbook, which consists of broad texts from STEM domains, its benefits are reflected in improved performance on STEM-related benchmarks like BBH, Math, and Code. Similarly, some datasets like Reddit are deemed to be generally helpful for improving the model's world knowledge and conversational abilities and are not limited to specific domain evaluations.
>
> > I notice that the training of FlexOlmo requires substantial computational resources in Section 4.4. In practical decentralized training scenarios, however, data owners may not have access to such extensive resources for continual training. This limitation should be discussed.
>
> The substantial computational resources mentioned in Section 4.4 (512 H100s for 3 days) are for training the base public model on 1T tokens. Training the individual local experts requires significantly less compute (e.g., 50B tokens for one expert).
> Our framework is applicable to cross-silo federated learning, where clients are typically organizations with access to reasonable compute resources. Furthermore, as we will discuss in the limitations section, future work can explore extending our model to accommodate clients with varying compute budgets by allowing them to train different-sized local experts.
>
> > From experiments, the majority of FlexOlmo's performance gains appear to come from "Math2". If the math domain is excluded, does FlexOlmo still outperform BTM and BTX?
>
> Yes, it does. After excluding the Math2 results, FlexOlmo's average score is still better than the baselines. The average scores are: FlexOlmo (47.4), BTM (46.5), and BTX (43.3).
>
> > What is the motivation behind processing and filtering the original Reddit dataset? Why not directly use the same version adopted in previous studies?
>
> In the case of the Reddit data, we drew on data from a parallel line of work, which was motivated by the observation that prior versions of Reddit data showed limited performance gains on downstream tasks, particularly on knowledge-based QA tasks. The hypothesis of that work was that Reddit's valuable, specialized knowledge could be better leveraged by more targeted filtering and data refinement. The process to arrive at the final data involved (1) constructing thread contexts inspired by QA structure, (2) filtering for high-quality, academically relevant subreddits, and (3) rewriting the content to reduce noise and match standard QA formats. The effectiveness of this approach was validated experimentally: when continuing pretraining on OLMo 7B, the rewritten Reddit data yielded substantial improvements over simply using the filtered data – **MMLU performance improved from 0.62 to 0.66, and MC9 improved from 0.74 to 0.76.** This Reddit data is released to facilitate future research.
>
> > What is the rationale for selecting these eight domains? Are there any references or prior work supporting the inclusion of domains such as "Creative Writing" or "Reddit"? Furthermore, is there any semantic relationship or overlap among these domains? The core contribution and motivation of the constructed dataset FlexMix is unclear. And I suggest the authors explicitly list their main contributions in the Introduction section.
>
> The seven domains in FlexMix were chosen to represent common, realistic scenarios where data access is restricted. For example, "Creative Writing" serves as a proxy for copyrighted materials like fiction and books, and "News" data represents copyrighted news data, which correspond to real-world lawsuits where major publishers like The New York Times [2] have recently restricted the use of their raw news corpus for pretraining LMs and Anthropic [3] face legal charges for using pirated books for pretraining. "Reddit" in FlexMix represents online discussion forums, which currently face tangible access restrictions; notably, this data is not downloadable following Reddit's 2023 access policy change [1]. These domains exemplify cases where data is not openly available due to its proprietary nature or user privacy concerns. While semantic overlap can exist between domains (e.g., textbooks containing math text), this complexity reflects real-world data.
>
> FlexMix provides a valuable testbed for advancing research in privacy-conscious and data-flexible model pretraining. On this benchmark, we show that FlexOlmo can effectively leverage these restricted sets to produce a better LM.
> Finally, thank you for the suggestion regarding the contribution list; we will explicitly list our main contributions in the Introduction section of the final paper.
>
> We hope our responses have addressed the reviewer's concerns. Please let us know if you may have any questions.
>
> [1] https://the-decoder.com/reddit-ends-its-role-as-a-free-ai-training-data-goldmine/
>
> [2] https://www.popsci.com/technology/nyt-generative-ai/
>
> [3] https://fortune.com/2025/07/28/a-copyright-lawsuit-over-pirated-books-could-result-in-business-ending-damages-for-anthropic/

---

> > ### Comment · Reviewer_F5CK · 2025-08-01
> >
> > Thank you for your efforts in addressing my concerns.
> > I acknowledge that at least half of my concerns — specifically weaknesses 3–6 — have been satisfactorily resolved.
> >
> > However, regarding weakness 1, I am still unclear about the parameter update, particularly the second step mentioned in your rebuttal. Could you clarify how the parameter is updated jointly? If possible, please provide some mathematical formulations to illustrate this.
> >
> > Additionally, please ensure that the relevant revisions are incorporated into your revised paper.

---

> ### Author Response · Authors · 2025-08-01
>
> Thank you for your feedback! We will ensure all the revisions are incorporated into the final paper.
>
> Regarding weakness 1, here is our clarification:
>
> For the 2nd step of local training, FlexOlmo uses a Mixture-of-Experts model with two experts: a trainable **local expert** ($M_i$) and a frozen **public expert** ($M_{pub}$). Given a loss function $\mathcal{L}$ calculated on the local dataset $D_i$, the parameter updates via gradient descent are as follows:
>
> ### Trainable Parameters
> The local expert's parameters ($\theta_i$) and its router embedding ($r_i$) are updated.
>
> $$\theta_i \leftarrow \theta_i - \eta \frac{\partial \mathcal{L}}{\partial \theta_i}$$
>
> $$r_i \leftarrow r_i - \eta \frac{\partial \mathcal{L}}{\partial r_i}$$
>
> ### Frozen Parameters
> The public expert's parameters ($\theta_{pub}$), its router embedding ($r_{pub}$), and shared attention layers remain unchanged. The public expert and its router embedding are shared across different local training, serving as a common anchor that ensures all local experts learn to coordinate with the same reference model.
>
> We are happy to answer any further questions the reviewer may have and respectfully ask that our revisions be considered in your final evaluation.

---

> > ### Comment · Reviewer_F5CK · 2025-08-01
> >
> > Thank you for the further clarification. If I understand correctly, both the router and the trainable expert models are updated through the same loss function, right?
> > I think I have a clearer picture about the training process.
> >
> > I will raise my score accordingly. Best of luck with the acceptance!

---

> > > ### Author Response · Authors · 2025-08-01
> > >
> > > Yeah, both the router and the trainable expert models are updated through the same loss function. We will clarify it in the final version of the paper. Thank you!

---

### Official Review · Reviewer_HG3R · 2025-06-28

**Clarity:** 4
**Significance:** 4
**Originality:** 4
**Rating:** 6
**Confidence:** 5

**Summary:**

FLEXOLMO is a novel class of Language Models (LMs) designed to overcome the significant challenges associated with centralized data access and control in traditional LM training. Unlike conventional methods that require aggregating all data centrally and making a one-time decision about data inclusion, FLEXOLMO enables distributed training without data sharing and provides flexible, fine-grained control over data access during inference. FLEXOLMO employs a mixture-of-experts (MoE) architecture. In this design, individual "experts" (small feedforward networks or FFNs within each transformer block) are trained independently on private, locally maintained datasets. A shared "public model" (Mpub) trained on publicly available data serves as an anchor point. FLEXOLMO supports flexible inclusion or exclusion of experts and their associated data during inference with no further training. This provides strong guarantees for data opt-out. For example, the paper demonstrates that removing a "news expert" significantly reduces performance on news-related tasks but has minimal impact on unrelated tasks. This directly addresses societal impacts related to data access, intellectual property, and privacy regulations (like GDPR and HIPAA).

**Questions:**

1. While the breadth of evaluation is a strength, the absence of statistical significance measures (like error bars or confidence intervals) for the reported performance metrics (e.g., in Table 1 and Table 2) is a notable limitation.
2. authors should clarify: How inference-time FLOPs/latency scale as the total number of experts (n) increases and as the number of active experts (k) varies (e.g., does activating only 4 out of 8 experts significantly reduce compute compared to a dense model of equivalent capacity?).
Clarify, how the overall training cost scales when adding new experts? While experts are trained independently, there might be implications for the shared anchor or router components.

**Ethical Concerns:**

["NO or VERY MINOR ethics concerns only"]

**Limitations:**

Yes

**Quality:**

4

**Strengths And Weaknesses:**

The paper is well-organized, with clear sections detailing the problem, proposed architecture, training algorithm, experimental setup, and results.
FLEXOLMO employs a mixture-of-experts (MoE) architecture where individual experts are trained independently on private datasets, a stark contrast to traditional centralized training. A key innovation is the nonparametric routing mechanism, which allows experts to be integrated without any joint training, a major challenge in decentralized settings. This is further refined by a learnable negative bias term for each independently trained expert, which helps the router narrow decision boundaries and improves generalization. The model was extensively evaluated on 31 diverse downstream tasks across general-purpose and domain-specific categories.


weakness -
 The paper does not report error bars for experimental results due to "prohibitive expense". While a large suite of benchmarks is used to mitigate this, the absence of statistical significance measures is a noted limitation. While compute resources are reported, a formal analysis of how FLEXOLMO's computational complexity scales with the number of models (N) and objectives (K) is not provided, which could be useful for practitioners.


◦

---

> ### Author Rebuttal · Authors · 2025-07-31
>
> > The paper does not report error bars for experimental results due to "prohibitive expense". While a large suite of benchmarks is used to mitigate this, the absence of statistical significance measures is a noted limitation.
>
> We thank the reviewer for noting this. The costs associated with training the models unfortunately make it impossible to perform multiple runs for each experiment, which is required for reporting error bars. We acknowledge this is a limitation and will include this discussion in the paper.
>
> > While compute resources are reported, a formal analysis of how FLEXOLMO's computational complexity scales with the number of models (N) and objectives (K) is not provided, which could be useful for practitioners. How inference-time FLOPs/latency scale as the total number of experts (n) increases and as the number of active experts (k) varies (e.g., does activating only 4 out of 8 experts significantly reduce compute compared to a dense model of equivalent capacity?). Clarify how the overall training cost scales when adding new experts? While experts are trained independently, there might be implications for the shared anchor or router components.
>
> - Training Cost: The cost of adding a new expert is independent of the total number of existing experts and only depends on the local expert size. Our model can scale efficiently in training as there is no joint training of all experts.
>
> - Inference Cost: Similar to standard MoE, Inference costs are dependent on the number of active experts (k), not the total number of experts in the model. This means that activating only 4 out of 8 experts significantly reduces compute compared to a dense model of equivalent capacity.
>
> We believe the FlexOLMo approach is very appealing in this regard, and we will include a detailed discussion regarding computational complexity scaling in the final version of the paper.
>
> We hope our responses have addressed the reviewer's concerns. Please let us know if you may have any questions.

---

> > ### Comment · Reviewer_HG3R · 2025-08-01
> >
> > Thank you for addressing my comment. I am happy with the response. Please make sure that you include the discussion regarding computational complexity scaling in the final version.

---

### Official Review · Reviewer_7oic · 2025-07-02

**Clarity:** 2
**Significance:** 3
**Originality:** 3
**Rating:** 5
**Confidence:** 4

**Summary:**

This paper presents FlexOLMo, a novel LLM specifically designed to address two key real-world constraints: distributed training without data sharing, and data-flexible inference. The model employs a MoE design, where each expert is trained independently and asynchronously on private data using FlexMix where each experts are later integrated via a nonparametric router, leveraging  as an open-source anchor model, without requiring any joint training. FlexOLMO outperforms both prior model merging baselines and individual experts across a diverse set of benchmarks.

**Questions:**

1. Could the authors elaborate on what they mean by "fine-grained control of data access during inference"? How exactly does this mechanism work, and what are the specific benefits? Is there any quantitative analysis as support during evaluations that demonstrates the advantages of this approach?
2. Why does training on math-mix experts improve code performance? What is the underlying connection between mathematical reasoning and coding capabilities?
3. What is the specific accuracy of the router? For example, when the router classifier is replaced and ground truth labels are directly provided for each routing operation, how is FlexOlmo's accuracy affected?
4. How can the combined model outperform individual expert X that was specifically trained on domain X (Table 1, Math-mix increase Code4 accuracy by 14.7%)? What is the intuition behind this?

**Ethical Concerns:**

["NO or VERY MINOR ethics concerns only"]

**Final Justification:**

I believe this work presents a novel and promising direction that should be accepted. My previous concerns, including those regarding the choice of CPT, have been resolved. Assuming the revised manuscript will address the motivation for CPT, challenges with SFT and RL, and discuss cross-domain generalization, I am increasing my scores for the significance, my confidence in this work, and the overall score.

**Limitations:**

yes

**Paper Formatting Concerns:**

I don't notice any paper formatting concerns.

**Quality:**

3

**Strengths And Weaknesses:**

### Strengths
 - Authors propose a novel MoE training framework where each expert trains independently before being merged and enabled to communicate with one another, which shows promising future research due to its adaptability.
- They conduct comprehensive evaluations across 31 different taks, along with detailed ablation studies examining factors such as expert count effects and routing pattern analysis.
- The proposed approach is both practical and efficient for real-world applications, outperforms some prior merging methods.

### Weaknesses
- The main limitation of the proposed work is its exclusive focus on large-scale datasets and continued pre-training. The scope of FlexOlmo would be more comprehensive if it could demonstrate SFT and RL based training for individual experts across different domains. The motivation for relying solely on large-scale datasets, specifically creating FlexMix and continued pre-training remains unclear.

- As far as I understand, experiments are conducted using only one public model in 7B scale. To validate the scalability of the proposed method, it would be more reliable to include two other open-source models such as Llama-based or Qwen-based architectures. Moreover, it would be interesting to examine the parameter scalability of FlexOlmo by testing whether the observations hold across varying model sizes (e.g., Qwen-3B, Qwen-7B, Qwen-32B).

- Authors mention that FlexOlmo trains each expert independently compared to joint training, which is promising for scalability and adaptability to new tasks and domains. However, it remains unclear how this coordination is achieved. If experts are trained in complete independence, how do they learn to coordinate effectively? Is there a specific optimization mechanism implemented during independent training to facilitate this coordination?

---

> ### Author Rebuttal · Authors · 2025-07-31
>
> We thank the reviewer for their time and constructive feedback. We appreciate the acknowledgment that our method is both practical and efficient for real-world applications. We address each of your comments below.
>
> > The main limitation of the proposed work is its exclusive focus on large-scale datasets and continued pre-training. The scope of FlexOlmo would be more comprehensive if it could demonstrate SFT and RL based training for individual experts across different domains. The motivation for relying solely on large-scale datasets, specifically creating FlexMix and continued pre-training remains unclear.
>
> We agree that extending the FlexOlmo framework to SFT and RL is a valuable direction for future research. Our choice to focus on contined pre-training is motivated by pressing, real-world cases where data owners possess large amounts of restricted, unstructured data rather than instruction-tuning or preference datasets. For example, the Reddit data [1] in FlexMix represents real-world unstructured data that now comes with access restrictions. Similarly, major publishers like The New York Times [2] have recently restricted the use of their raw news corpus for pretraining LMs. Anthropic [3] faces legal charges for using pirated books like libgen for pretraining. These access restrictions create a critical bottleneck: valuable data for improving LM is locked away.
> FlexOlmo addresses this problem by giving data owners more control over their data in pretraining. While we believe FlexOlmo's modular training approach would generalize effectively to SFT and RL, exploring that is a natural next step that remains beyond the scope of our current paper and opens up technical considerations that would require extensive experimentation. We will include a discussion on extending our work to SFT and RL in the future work section. Thanks for the suggestions.
>
> > Extending to Other Open-Source Architectures
>
> We note that we cannot use Llama or Qwen models directly, as our research requires pre-training a model from scratch to ensure no overlap between the pre-training data and high-risk data added later. While it would be possible to pre-train models based on Llama- or Qwen-based architectures, it is computationally costly, requiring 512 H100 GPUs for three days, which exceeded our budget. We leave the exploration of FlexOlmo under alternative architectures for future work.
>
>
> > If experts are trained in complete independence, how do they learn to coordinate effectively
>
> This is related to the core technical innovation of FlexOlmo. Our training method is explicitly designed to enable coordination of experts without joint training. We achieve this through two key mechanisms:
>
> 1. Coordinated Training with an Anchor Model: Instead of joint training, each data owner trains their expert alongside a frozen copy of the public model. **This public model acts as a shared "anchor,"** ensuring that all independently trained experts learn to operate within a common functional space and can coordinate effectively when merged.
>
> 2. Domain-Informed Router Initialization: The router, which directs tokens to the appropriate experts, is initialized using domain-specific embeddings derived from each expert's training data. These embeddings encode useful information on how to differentiate between the experts, providing a strong starting point for the router.
>
> > Could the authors elaborate on what they mean by "fine-grained control of data access during inference"? How exactly does this mechanism work, and what are the specific benefits? Is there any quantitative analysis as support during evaluations that demonstrates the advantages of this approach?
>
> “Fine-grained control” means that a single FlexOlmo model can deploy different combinations of experts (and their associated data) to different users based on specific access criteria such as permissions, access policies, or the user’s own preferences. Here is a concrete example – suppose a major newspaper contributed an expert. They could restrict access so that only users with an active subscription to that paper have the option to activate that expert, while others cannot. We quantitatively demonstrate the effect of data opt-out in section 5, showing that deactivating the News expert successfully removes its influence on in-domain tasks while having minimal impact on unrelated tasks.
>
>
> > Why the math expert improves coding performance
>
> The improvement in coding performance from the Math expert is likely due to cross-domain generalization of reasoning abilities. The structured  step-by-step problem-solving inherent in math text appears to enhance the model's underlying reasoning capabilities. This phenomenon has been observed in prior work; for example, [4] showed that training on mathematical data can significantly boost performance on coding benchmarks and vice versa.
>
>
> > What is the specific accuracy of the router? How can the combined model outperform individual expert X that was specifically trained on domain X
>
> A traditional accuracy metric is not applicable because our router operates at the **token level, not the sequence level**. There is no single ground-truth expert label for a token. This token-level routing can dynamically select the most appropriate expert for each token in the input sequence. For example, consider an example that contains both math concepts and coding elements. With token-level routing, our model can route the math tokens to the math expert while directing the coding-related tokens to the coding expert. This token-level routing allows the model to leverage each expert’s expertise precisely where it's needed within a single example, rather than being constrained to use one expert for the entire example.
>
> We hope our responses have addressed the reviewer's concerns. Please let us know if you may have any questions.
>
> [1] https://the-decoder.com/reddit-ends-its-role-as-a-free-ai-training-data-goldmine/
>
> [2] https://www.popsci.com/technology/nyt-generative-ai/
>
> [3] https://fortune.com/2025/07/28/a-copyright-lawsuit-over-pirated-books-could-result-in-business-ending-damages-for-anthropic/
>
> [4] Magistral

---

> > ### Comment · Reviewer_7oic · 2025-08-01
> >
> > Thanks authors for their point-by-point response, my issues regarding w3 and additional questions are resolved. Although 2/3 of the weaknesses I highlighted were addressed as future work (w1, w2), I understand the limitations regarding the second point (w2), assuming the computational issues discussed in the latter version of the paper. However, I still have concerns about w1.
> >
> > Specifically, I agree that exploring RL could reasonably be left for future work, but I remain unclear about the SFT part. Should we understand that, in its current form, FlexOlmo is essentially just a sentence completion base model which does not know how to follow instructions (assuming no instruction data in pre-training) but able perform multiple choice-qa tasks like MMLU that well, as shown in Table 1? This seems unrealistic to me but I am willing to increase my score if authors elaborate more on this CPT/SFT discussion and corresponding performance on the reported downstream tasks. On the other hand, does the motivation for CPT come from the challenge of converting that type of "unstructured data" (i.e., large and noisy) into instruction format for SFT, or is it due to something else?

---

> ### Author Response · Authors · 2025-08-01
>
> Thank you for your constructive feedback! We will incorporate a detailed discussion of SFT, RL, and CPT in the limitations section.
>
> We agree that CPT, SFT, and RL each serve different purposes in the model development lifecycle. The choice between these approaches depends primarily on the format and characteristics of the restricted data:
>
> - **Supervised Fine-Tuning (SFT)** is effective when the restricted data itself is in instruction format. For example, if a data owner possesses extensive user-model conversation histories, SFT can directly leverage this structured interaction data.
>
> - **Continued Pre-Training (CPT)** is designed for absorbing domain-specific knowledge and is well-suited for unstructured data. This approach aligns with our current setup, where the restricted data consists of plain text such as news and books. Your suggestion to convert unstructured data into instruction format is a promising direction; it introduces technical considerations that would merit extensive experimentation.
>
> Fundamentally, **CPT and SFT are similar in nature, sharing the same next-token prediction objective, differing mainly in their input data format.**  Furthermore, prior research [1, 2] suggests that core model capabilities are primarily from from pretraining, making CPT the more effective method for integrating broad domain-specific knowledge. To make the model interact using chat format, a small, high-quality SFT dataset is often sufficient to align the model.
>
> We acknowledge that a CPT-only model is not production-ready for end users. To enable instruction-following capabilities, people can easily perform SFT with a small number of examples on top of FlexOlmo (which is a base model) or even use in-context learning [3] to replace SFT/alignment. To validate the feasibility of SFT, we conducted a preliminary experiment. We performed SFT on our FlexOlmo model and compared it with the OLMo-2 instruction-following version. As shown in the table below, our initial results demonstrate that our model outperforms the baseline OLMo-2-7B on several key benchmarks. Through testing the fine-tuned models, we confirm that they possess instruction-following capabilities:
>
> | Model        | Codex HumanEval+ | IFEval | Math : Flex |
> |--------------|------------------|--------|-------------|
> | FlexOlmo SFT | **71.7**          | 65.6   | **37.2**     |
> | OLMo-2 SFT   | 37.2             | 65.6   | 25.9        |
>
> Although further hyperparameter optimizations may require additional time, this remains an active area of exploration. We will clarify the complete method and its preliminary validation in the final version of the paper.
>
> Thank you again for this suggestion. We are happy to answer any further questions the reviewer may have and respectfully ask that our revisions be considered in your final evaluation.
>
> [1] LIMA: Less Is More for Alignment
> [2] The False Promise of Imitating Proprietary LLMs
> [3] The Unlocking Spell on Base LLMs: Rethinking Alignment via In-Context Learning

---

> > ### Author Response · Authors · 2025-08-04
> >
> > Thank you for the feedback and suggestion. Please let us know if you have any further questions. We would be happy to address them.

---

> > > ### Comment · Reviewer_7oic · 2025-08-04
> > >
> > > Special thanks to authors for providing further explanations regarding the definitions of these post-training methods (which we were already aligned) and present an additional experiment demonstrating the SFT performance. However, this raises further questions like which data was used for the SFT experiments, how was this data generated, etc.? I believe my main concerns in my previous response remain unaddressed and a more specific response to those would be helpful:
> > >
> > > - To re-iterate, specifically, I am still unclear about the reported accuracy improvements on both knowledge and reasoning tasks during evaluation. While I agree that "CPT is for knowledge injection" and "SFT is for instruction following capabilities", the reported benchmarks require both knowledge stored in the model parameters and strong instruction-following abilities. To achieve the observed delta accuracy improvements among all benchmarks, with CPT-only applied settings, still seems unrealistic without further clarification from authors. Approach is certainly novel, but the evaluation details/discussions need to be more clear in that sense; especially in these "evaluation crisis" days with non-reproducable results. As it stands, future readers may be left with the same confusion I have.
> > >
> > > - Additionally, as I previously asked for: is the primary motivation for CPT-only approach for FlexOlmo comes from the difficulty of converting large and unstructured data to instruction format, or are there other reasons I miss? I would appreciate further elaboration on this point, as it will be again crucial for third-party users of FlexOlmo.

---

> ### Author Response · Authors · 2025-08-04
>
> Thank you for your continued feedback. We appreciate your thorough review and would like to address your specific concerns:
>
> We would like to clarify **FlexOlmo presents a new pretraining recipe for building a pretrained base LM, developed from scratch on unstructured data** in a manner similar to OLMo [3], Llama [2] and DeepSeekMoE [4]. Our main experiments are designed to demonstrate the effectiveness of this new pretraining approach. This pretrained model can then be aligned for instruction-following through standard post-training procedures using a small, high-quality, general-domain instruction tuning dataset (as shown in our previous response). As shown in the FLOPs-controlled setup in Table 1, FlexOlmo proves to be a more effective base model compared to one trained with a standard MoE pretraining recipe [4].
> With that context, we would like to address your specific points:
>
> > **Regarding SFT Experimental Details:**
>
> For our Supervised Fine-Tuning experiments in the previous response, we used a general-domain instruction-tuning dataset from Tulu 3 [1]. The primary purpose of this experiment was to validate that FlexOlmo can be successfully aligned with user instructions through standard post-training procedures, thereby demonstrating the model's compatibility with existing alignment workflows.
>
> > **Is the primary motivation for CPT-only approach for FlexOlmo comes from the difficulty of converting large and unstructured data to instruction format, or are there other reasons I miss?**
>
> The decision was not driven by difficulty of data conversion, but our research objective: developing a better pretraining recipe with new functionalities (decentralized training, data opt-in and out) for base LMs. We conduct controlled comparison against other pretraining approaches on the same data  in the main experiments. Instruction-following capabilities can be added later by applying standard post-training techniques using curated instruction datasets from general domains like Tulu 3[1].
>
> > **The evaluation details/discussions need to be more clear in that sense; especially in these "evaluation crisis" days with non-reproducable results.**
>
> We made all our models, training code, and evaluation scripts publicly available. Both our FlexOlmo models and baseline results can be fully reproduced using our released codebase. Our experimental setup ensures baseline pretraining approaches and our method are evaluated under identical resource constraints and using the same datasets.
>
>
> [1] Tulu 3: Pushing Frontiers in Open Language Model Post-Training
>
> [2] LLaMA: Open and Efficient Foundation Language Models
>
> [3] OLMo: Accelerating the Science of Language Models
>
> [4] DeepSeekMoE: Towards Ultimate Expert Specialization in Mixture-of-Experts Language Models

---

> > ### Comment · Reviewer_7oic · 2025-08-05
> >
> > Thank you to the authors for their response, my concerns regarding the CPT details have been now resolved.
> >
> > Assuming the paper will address the motivation for CPT and challenges with SFT/RL, and add the discussion around cross-domain generalizations in revised manuscript; I believe this work presents a new direction and promising approach which should be accepted. Thus, I am increasing my score accordingly.

---

### Official Review · Reviewer_JVHX · 2025-07-08

**Clarity:** 3
**Significance:** 4
**Originality:** 3
**Rating:** 6
**Confidence:** 4

**Summary:**

This paper introduces FlexOLMo, a novel class of open language models that support both distributed training without data sharing and data-flexible inference. At the core of the approach is a Mixture-of-Experts (MoE) architecture where each expert is trained independently on a private dataset using a shared public model as the anchor. Experts are merged post-training via a nonparametric router constructed from domain-informed embeddings, avoiding any joint training. The system allows opt-in/opt-out control at inference time, addressing data sovereignty, licensing, and privacy concerns. The authors demonstrate FlexOLMo utility using a curated dataset (FLEXMIX) composed of a public mix and seven restricted domains. They show that FLEXOLMO consistently outperforms strong baselines such as model soups, BTX, BTM, and prompt-based, while also supporting modular extensibility and strict data-use guarantees.

**Questions:**

1. Could the authors provide more empirical evidence or analysis on how well the router generalizes?

2. What is the hypothesized reason for FlexOlmo underperforming on Code4?

3. Can the proposed framework accommodate settings where different data providers use different model architectures for their local experts? If not, what modifications would be needed to enable this?

4. Why was federated learning not included as a baseline, given that it is a direct solution to the same data-sharing constraint? Additionally, have the authors considered comparisons to model routing methods such as:
[1] Muqeeth, Mohammed, et al. "Learning to route among specialized experts for zero-shot generalization." ICML 2024.

**Ethical Concerns:**

["NO or VERY MINOR ethics concerns only"]

**Final Justification:**

I support the paper’s acceptance, as it presents a promising direction that could benefit the community.

**Limitations:**

The authors have not included a dedicated section or explicit discussion of the limitations of their work. While the paper mentions challenges like underperformance in coding tasks (Tables 1 and 2), it does not systematically address limitations such as potential scalability issues and computational costs. Additionally, there is no discussion of potential negative societal impacts, such as biases in the restricted datasets or ethical concerns related to data opt-out guarantees.

**Paper Formatting Concerns:**

No major formatting issues noted.

**Quality:**

3

**Strengths And Weaknesses:**

**Strengths:**
1. The paper tackles two highly important and challenging problems in current language model development: data sharing constraints and data use constraints. These issues are increasingly central due to legal, privacy, and IP considerations, and FlexOLMo offers a meaningful solution.

2. The proposed combination of modular, distributed training and data-flexible inference via a nonparametric MoE framework is both practical and elegant. The authors demonstrate solid performance improvements across a wide range of tasks, outperforming competitive baselines.

**Weaknesses:**

1. While the paper evaluates against several baselines, it omits comparison to relevant approaches based on federated learning or other model merging methods, which also address the same data restrictions problem. For example, works like Muqeeth et al. (ICML 2024) propose routing among specialized experts in a related context and could serve as valuable baselines.

2. While the architecture is well-motivated and practically useful, the technical contributions over prior MoE-based approaches appear incremental rather than fundamentally novel.

---

> ### Author Rebuttal · Authors · 2025-07-31
>
> Thank you for the thoughtful review and for recognizing our work as a meaningful approach to the critical challenge of data constraints in language model development. We address each of your comments below and have incorporated new experiments and analyses:
>
> > Why was federated learning not included as a baseline, given that it is a direct solution to the same data-sharing constraint? Additionally, have the authors considered comparisons to model routing methods such as: [1] Muqeeth, Mohammed, et al. "Learning to route among specialized experts for zero-shot generalization." ICML 2024.
>
> **Comparison to Muqeeth et al. (ICML 2024):** Thanks for the suggestion. We have adapted the SMEAR method from Muqeeth et al. to our experimental setup. As shown in the table below, our proposed method outperforms SMEAR across all evaluation categories. We will include this result in the final version of the paper:
>
> | Model | MC9 | Gen5 | MMLU | MMLU Pro | AGI Eval | BBH | Math2 | Code4 | Avg. |
> |-------|-----|------|------|----------|----------|-----|-------|-------|------|
> | SMEAR (Muqeeth et al.) | 69.6 | 56.3 | 58.6 | 30.5 | 45.4 | 43.5 | 18.5 | 14.8 | 42.2 |
> | FlexOlmo | 71.0 | 59.8 | 59.9 | 30.8 | 45.8 | 47.1 | 50.7 | 17.3 | 47.8 |
>
> **On Federated Learning (FL):** We experimented extensively with existing FL algorithms (e.g., FedAvg, FedProx) using the NVFlare framework (https://github.com/NVIDIA/NVFlare). However, we encountered substantial engineering challenges in scaling these methods to our setting (training models with up to 37 billion parameters), primarily due to the synchronization required in FL. Existing FL methods are not yet supported for training at this scale. Additionally, FL methods do not support data opt-out at inference time, as they are designed to produce a single, monolithic model.  We will add discussion of these challenges to the paper.
>
> Our paper includes comparisons to four strong model merging baselines, including BTM and BTX. We believe these, along with the new SMEAR results above, provide a comprehensive comparison. We are happy to include additional baselines in the final version if the reviewer has additional specific suggestions.
>
>
>
> > The technical contributions over prior MoE-based approaches appear incremental rather than fundamentally novel.
>
> We respectively disagree with this assessment. While Flexolmo borrows the MoE architecture, its core contribution is a fundamentally new training method that enables capabilities not supported by standard MoEs. Standard MoE models are trained end-to-end with joint access to all data. In contrast, our method introduces a new, coordination-based training algorithm that enables two key features: (1) distributed training without data sharing and (2) opting in and out of subsets of data during inference. These features are important for leveraging restricted datasets where centralized data pooling is not feasible.
>
> Additionally, we chose to keep the standard MoE architecture in Flexolmo to ensure our method can be easily integrated into existing large-scale pretraining LM pipelines.
>
>
> >  What is the hypothesized reason for FlexOlmo underperforming on Code4?
>
> We attribute this to suboptimal routing. Ideally, coding tasks should be mostly handled by the code expert, but current routing sometimes directs them to other experts. To support this, we analyzed the router’s behavior on coding benchmarks and found that ~23% of tokens are routed to the Textbook and Math experts. We will include this analysis in the final version of the paper to accelerate future improvements.
>
>
> > Can the proposed framework accommodate settings where different data providers use different model architectures for their local experts?
>
> The overall architecture of the local experts needs to be the same as the anchor model (e.g., same positional encoding). Varying the FFN’s hidden dimension is feasible for different local experts as long as the output vectors of the experts have the same dimension. The standard FFN expert has two linear layers: an up-projection to an intermediate dimension and a down-projection back to the output dimension. This intermediate dimension, and thus the FFN expert size, can vary based on a provider's local data size and compute constraints.
>
> > Limitation sections
>
> We thank the reviewer for this suggestion. Below is a draft of a limitations section that we will include in the final version of the paper:
>
> **Data privacy and extraction risks.** A primary concern with sharing model weights is the potential for data leakage. As we quantify in our data extraction analysis (Section 5.3), while the rate of extractable training data from an expert's weights is low, it may still be too high for high-stakes domains like healthcare or finance. A direction for future work is to enable data owners to apply differential privacy (DP) training during their local expert training. Since DP is orthogonal to the FlexOlmo architecture, each contributor could independently decide whether to adopt it, providing formal privacy guarantees.
>
> **Scaling and model size.** The FlexOlmo architecture scales by adding new experts, which linearly increases the total number of model parameters. While the sparse activation could keep the number of active parameters constant during inference, the total model size can become substantial as more experts are integrated. Future work could explore techniques like expert distillation or pruning to manage model size.
>
> **Bias and ethics concerns of FlexOlmo.** While the modularity of FlexOlmo provides powerful data control, it also introduces risks that require careful consideration. One concern is the potential for malicious use, where an adversary could intentionally contribute a low-quality expert trained on misinformation. Integrating this low-quality expert could compromise the integrity and safety of the entire model.  Similarly, low-quality experts could compromise the model even without any ill intent. Addressing such scenarios would be valuable future work. Furthermore, the framework's guarantee of data owner control raises governance and ethical questions. For instance, an unethical data owner could engage in discriminatory gatekeeping: deciding to withhold a valuable expert (e.g., in medicine or finance) from certain users.
>
> We hope our response and the additional results have addressed the reviewer's concerns. Please let us know if you may have any questions.

---

> > ### Comment · Reviewer_JVHX · 2025-08-05
> > **Good Rebuttal**
> >
> > Thank you to the authors for the additional experimental results and clarifications. I have accordingly raised my score and appreciate the improved clarity and demonstrated performance. I support the paper’s acceptance, as it presents a promising direction that could benefit the community.
> >
> > I understand that comparing against federated learning methods would require substantial engineering effort, and I would be interested in future studies that compare the proposed method with approaches that use periodic model aggregation during training (the key feature of federated learning) rather than only with one-shot model-merging methods after training. Overall, the proposed model and datasets are compelling, particularly in the support for data opt-out, and can attract interest from a broad NeurIPS audience.

---

> ### Author Response · Authors · 2025-08-04
>
> Thank you for the feedback and suggestion. Please let us know if you have any further questions. We would be happy to address them.

---

### Decision · Program_Chairs · 2025-09-17

**Decision:**

Accept (spotlight)

**Comment:**

The submission presents FlexOLMo, a modular mixture-of-experts (MoE) framework designed to enable distributed training without data sharing and flexible inference with expert opt-in/opt-out. The key contributions include a new training recipe where local experts are trained alongside a frozen public model anchor, and a nonparametric router that integrates experts without joint training. Evaluations on the curated FlexMix dataset and 31 downstream benchmarks show consistent gains over strong baselines, with notable improvements in restricted-data domains. Strengths highlighted by reviewers include the strong practical relevance of addressing data sovereignty and licensing constraints, thorough empirical evaluation, and modular extensibility. Reviewers also noted limitations (e.g., lack of federated learning comparisons, limited discussion of computational costs), but the rebuttal provided new comparisons, new clarification, and detailed explanations of technical designs. Overall, this is a strong technical paper and potentially has a broad impact in the community, so I recommend to accept this paper with spotlight.